# Cost-effectiveness analysis and budget impact of rivaroxaban compared with dalteparin in patients with cancer at risk of recurrent venous thromboembolism

Lisa A de Jong ![ORCID],[1] Annette W G van der Velden,[2] Marinus van Hulst,[3,4]
Maarten J Postma[4,5]

► Prepublication history and supplemental material for this paper is available online. To view these files, please visit the journal online (http://dx.doi.org/10.1136/bmjopen-2020-039057).

For numbered affiliations see end of article.

**Correspondence to**
Dr Lisa A de Jong;
l.a.de.jong@rug.nl

## ABSTRACT

**Objectives** In the 'Comparison of an Oral Factor Xa Inhibitor With Low Molecular Weight Heparin in Patients With Cancer With Venous Thromboembolism' (SELECT-D) trial, rivaroxaban showed relatively low venous thromboembolism (VTE) recurrence but higher bleeding compared with dalteparin in patients with cancer. We aim to calculate the cost-effectiveness and budget impact of rivaroxaban compared with dalteparin in patients with cancer at risk of recurrent VTE.

**Setting** We built a Markov model to calculate the cost-effectiveness from a societal perspective over a 5-year time horizon for the Dutch healthcare setting.

**Participants** A hypothetical cohort of 1000 cancer patients with VTE entered the model with baseline characteristics based on the SELECT-D trial.

**Intervention** Six months of treatment with rivaroxaban (15 mg two times per day for first 3 weeks followed by 20 mg once daily) was compared with 6 months of treatment with dalteparin (200 IU/kg daily during month 1 followed by 150 IU/kg daily).

**Primary and secondary outcome measures** The primary outcome of the cost-effectiveness analysis was the incremental cost-effectiveness ratio (ICER). The robustness of the model was evaluated in probabilistic and univariate sensitivity analyses. A budget impact analysis was performed to calculate the total annual financial consequences for a societal perspective in the Netherlands.

**Results** In the base case and all scenarios, rivaroxaban were cost-saving while also slightly improving the patient's health, resulting in economically dominant ICERs. In the probabilistic sensitivity analysis, 77.8% and 98.7% of the simulations showed rivaroxaban to be cost-saving and more effective for a 5-year and 6-month time horizon, respectively. Rivaroxaban can save up to €11 326 763 (CI €5 164 254 to €17 363 231) in approximately 8000 cancer patients with VTE per year compared with dalteparin based on a 1-year time horizon.

**Conclusions** Treatment with rivaroxaban is economically dominant over dalteparin in patients with cancer at risk for recurrent VTE in the Netherlands. The use of rivaroxaban instead of dalteparin can save over €10 million per year, primarily driven by the difference in drug costs.

### Strengths and limitations of this study

► This analysis used sophisticated pharmacoeconomic modelling methods to conduct cost-effectiveness and budget impact analyses, presenting the economic impact on a patient and on a population level.
► Our model is based on timely, robust data from the important SELECT-D trial.
► Various additional scenarios were used to analyse the effect of different assumptions and clinical situations.
► We assumed a 6-month treatment duration for all patients, while in clinical practice the treatment duration may vary between patients.
► Due to lack of data, the productivity losses were not taken into account.

### INTRODUCTION

Venous thromboembolism (VTE), comprising both pulmonary embolism (PE) and deep vein thrombosis (DVT), is a major challenge in patients with cancer.[1] In addition to the characteristics of the cancer itself, cancer therapy (chemotherapy and cancer surgery) has effects on the patient's coagulation system and therefore increases the risk of VTE and bleeding.[2 3] VTE in patients with cancer can cause unnecessary hospitalisations, interruption or postponement of cancer treatment, and increased mortality, leading to decreased quality of life and increased costs.

VTE is treated with anticoagulation therapy, and this is continued as prophylaxis for recurrence over a longer period because of the high risk of recurrence during the first months after the initial VTE.[4] Vitamin K antagonists (VKAs) or direct oral anticoagulants (DOACs) are indicated for the treatment and prevention of VTE in the general population.[5] DOACs are a relatively new class of anticoagulants. Apixaban, dabigatran, edoxaban and rivaroxaban are the four DOACs that are

currently registered for the prevention of recurrent VTE in Europe. DOACs have a more beneficial efficacy/safety ratio, do not require routine measurements of the INR and show fewer food–drug and drug–drug interactions compared with VKAs.[6 7]

The guidelines recommend against the use of VKAs in patients with cancer because of potential drug interactions, liver dysfunction and malnutrition, all of which lead to fluctuations of the international normalised ratio (INR) and could result in negative patient outcomes.[8–11] Moreover, trials in cancer patients with VTE have shown that low molecular weight heparin (LMWH) is more effective in the prevention of recurrent VTE compared with VKA, without increasing bleeding risk.[12–14] Therefore, the guidelines recommend at least 6 months of therapeutic treatment with a daily subcutaneous injection of LMWH (eg, dalteparin) in patients with cancer.[8–11] However, recently, DOACs rivaroxaban and edoxaban were also added as treatment options for the prevention of recurrent VTE in patients with cancer. This recommendation

was based on the results from the 'Comparison of an Oral Factor Xa Inhibitor With Low Molecular Weight Heparin in Patients With Cancer With Venous Thromboembolism: Results of a Randomized Trial' (SELECT-D) and 'Edoxaban for the Treatment of Cancer-Associated Venous Thromboembolism' (HOKUSAI VTE Cancer) trials.[15 16]

The SELECT-D is a multicentre, randomised, clinical pilot trial in the UK; it is a head-to-head comparison of rivaroxaban and dalteparin in 406 patients with active cancer who had experienced a symptomatic PE, incidental PE or symptomatic DVT.[15] Incidental PEs are non-symptomatic PEs that are incidentally found during tumour imaging. The trial researchers found that rivaroxaban reduces the recurrence of VTE (6-month cumulative VTE recurrence rate: 4% vs 11%) at the cost of an increased risk of bleeding (6-month cumulative major bleeding (MB) rate: 6% vs 4%; 6-month cumulative clinically relevant non-major bleeding (CRNMB) rate: 13% vs 4%) compared with dalteparin. These results were

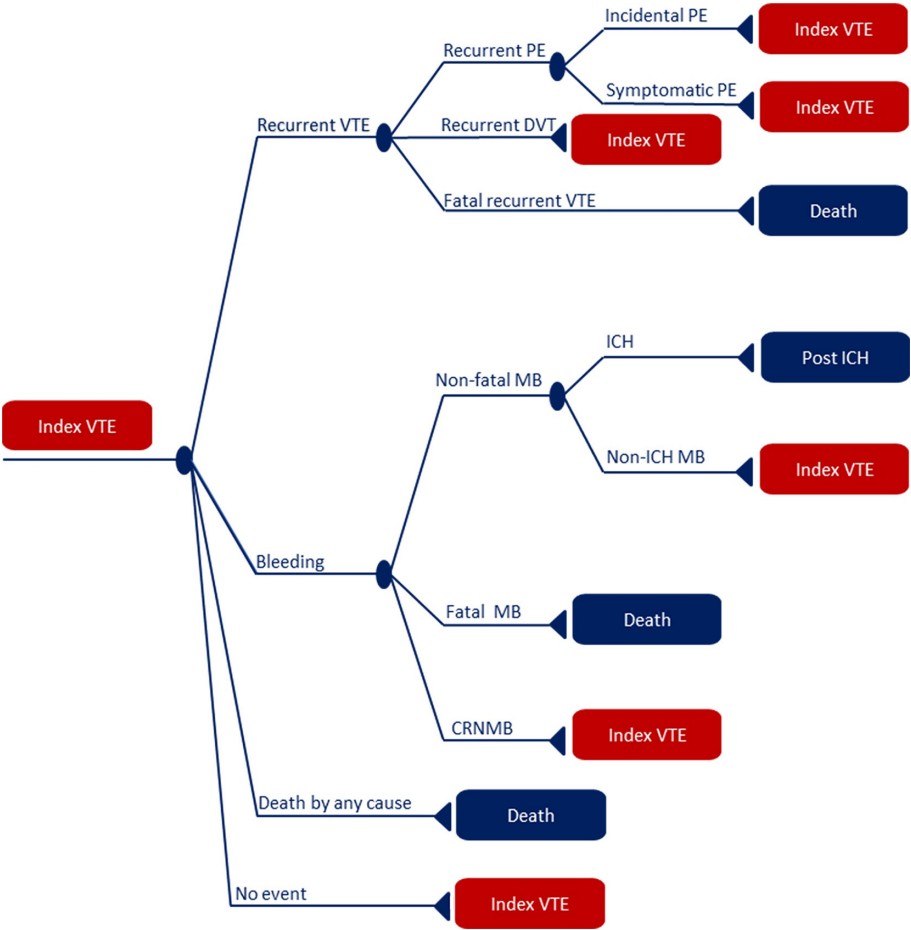

**Figure 1** Model outline. All patients enter the model in the 'Index VTE' state and move to other states on the occurrence of one of the following events: recurrent incidental PE, recurrent symptomatic PE, fatal recurrent VTE, recurrent DVT, ICH, non-ICH MB, fatal MB, CRNMB or death by any cause. The triangles represent the health state a patient will enter after an event. The blue squares are permanent states, in which a patient will remain until death while not being at risk for other events. The red squares represent a transient state: the patient will re-enter the model in the 'Index VTE' state. CRNMB, clinically relevant non-major bleeding; DVT, deep vein thrombosis; ICH, intracranial haemorrhage; MB, major bleeding; PE, pulmonary embolism; VTE, venous thromboembolism.

de Jong LA, *et al. BMJ Open* 2020;**10**:e039057. doi:10.1136/bmjopen-2020-039057

comparable with those of a large retrospective study by Streiff *et al*.[17]

Based on the results of these studies and the fact that DOACs can be orally administered (unlike the subcutaneously injected LMWHs), a greater utilisation of DOACs for VTE in patients with cancer might be expected. Since the introduction of DOACs, there has been an ongoing discussion about the economic impact of these drugs. To help guide this discussion and inform decision-making in this area, we designed and developed an economic model based on the SELECT-D trial to evaluate the cost-effectiveness and budget impact of rivaroxaban compared with dalteparin in patients with cancer at risk of recurrent VTE in the Netherlands.

## METHODS

The economic model comparing rivaroxaban with dalteparin was designed based on the SELECT-D trial,[15] since this study presented the most comprehensive results reflecting recurrent VTE and bleeding complications per event type (symptomatic PE, incidental PE and DVT) or severity (MB and CRNMB). The primary outcome of the cost-effectiveness analysis is the incremental cost-effectiveness ratio (ICER); this is calculated by dividing the incremental costs by the incremental health effects, expressed in quality adjusted life-years (QALYs). In accordance with Dutch costing guidelines for economic evaluations in healthcare, the ICER was calculated from a societal perspective, which incorporates direct and indirect costs both inside and outside the healthcare sector.[18] We performed sensitivity and scenario analyses to test the robustness of the model. Additionally, we conducted a budget impact analysis to reflect the annual financial consequences of the use of rivaroxaban in patients with cancer at risk of recurrent VTE in the Netherlands. The analysis was carried out in early 2019.

### Model outline

We developed a decision-tree-based Markov model using Microsoft Excel V.2016 to calculate the ICER. Figure 1 shows a schematic representation of the model, with the disease course being represented by separate health states. A hypothetical cohort of 1000 cancer patients with VTE entered the model with incidental PE, symptomatic PE or DVT, represented by the 'index VTE' health state. According to the guidelines, patients with incidental PE should be treated identically to those with symptomatic PE.[8 10] Patient characteristics were based on the SELECT-D trial protocol (table 1).[15] The SELECT-D population is representative for the Dutch cancer population, based on age, tumour type and gender distribution.[19] Patients move through various health states in the model during the follow-up time of 5 years. Five years was used because overall survival was assumed to be low after 5 years since the majority (58%) of the SELECT-D trial population had metastatic cancer.[15] We included the following health states in our model (see legend of figure 1 for abbreviations):

**Table 1** Patient characteristics of the hypothetical cohort of 1000 patients with cancer at risk of recurrent VTE

| Unit | Value | Reference |
|---|---|---|
| Age (years) | 67 | 15 |
| Proportion male | 53% | 15 |
| BMI (kg/m$^2$) | 25.6 | 15 |
| Type of cancer | | |
| Early or locally advanced cancer | 39% | 15 |
| Metastatic cancer | 58% | 15 |
| Haematologic malignancy | 2% | 15 |
| Distribution of PE and DVT | | |
| % index VTE that is symptomatic PE | 20% | 15 |
| % index VTE that is incidental PE | 53% | 15 |
| % index VTE that is DVT | 27% | 15 |

BMI, body mass index; DVT, deep vein thrombosis; PE, pulmonary embolism; VTE, venous thromboembolism

'recurrent incidental PE', 'recurrent symptomatic PE', 'fatal recurrent VTE', 'recurrent DVT', 'ICH', 'non-ICH MB', 'fatal MB', 'CRNMB', 'death by any cause' and 'no event'. Patients were assumed to remain in these states for one cycle, after which they moved back to the 'index VTE' state or the chronic, debilitating 'post-ICH' state, in which they remained until death without being at risk for any further complications. The cycle length was 1 month. Markov tunnel states (1 month post-VTE, 2 months post-VTE, …, 60 months post-VTE) were used to implement time dependency. These temporary states can only be visited once, which allows time-dependent future transitions, costs and health-related quality of life dependent on how long the patient has gone without a recurrent VTE event.[20] The chronic complications post-thrombotic syndrome (PTS) and chronic thromboembolic pulmonary hypertension (CTEPH) were modelled in the background. This means that PTS or CTEPH could occur at any time in the model, regardless of the health state the patient is in. Costs and health effects of these events were taken into account. However, only the severe cases of PTS were modelled, since the costs of minor PTS are considered negligible. For these chronic complications, we also used tunnel states since the risks of PTS and CTEPH were also time-dependent.

### Transition probabilities

Transition probabilities were used to calculate the number of patients in each health state per 1 month cycle. Online supplemental table S1 summarises all event rates presented in 6-month risks. The event rates were translated into monthly transition probabilities with the following formula:

$$P = 1 - \exp\{-rt\}$$

where P is the transition probability, r is the event rate and t is the cycle length (1 month).[20]

Event rates of recurrent VTE, MB and CRNMB in the first 6 months of treatment were based on the SELECT-D trial.[15] If patients did not experience a recurrent event during this period, anticoagulation treatment was discontinued. Recurrent VTE rates after treatment discontinuation were based on a retrospective study in patients with active cancer experiencing a VTE.[4] On the occurrence of a non-fatal recurrent VTE, patients were assigned to another 6-month treatment, with corresponding event rates. Bleeding risks after treatment discontinuation were based on the outcomes of the cancer population of the HOKUSAI VTE Cancer trial (which followed patients after edoxaban discontinuation for an additional 6 months) because these data are not reported for the SELECT-D trial.[16] The HOKUSAI VTE Cancer trial was also used to determine the distribution of ICH, non-ICH and fatal bleeding. The distributions among the different types of VTE (incidental PE, symptomatic PE, DVT and fatal PE) and MB (ICH, non-ICH, fatal MB) were calculated based on the total number of events in both arms (rivaroxaban and dalteparin) together and assumed it to be treatment independent, since the total number of events in the trials was low. The distributions of the types of VTE event were based on the number of recurrent VTE events observed in the SELECT-D trial in the lower extremities and pulmonary embolisms—other locations of VTE events (brachial, subclavian, jugular, renal plus inferior vena cava or the extrahepatic vein) were excluded.[15] Mortality rates (death by any cause) were based on Dutch cancer mortality data from the Netherlands Cancer Registry.[21] In the sensitivity analysis, all transition probabilities were varied over beta distributions. For percentages of the type of recurrent VTE and MB, a Dirichlet distribution was used in the sensitivity analysis. As recommended by the Dutch guidelines for economic evaluation of healthcare, the distributions were based on Briggs *et al*, who described the use of distributions around model input parameters (eg, distributions limited to positive values (costs) or even confined between 0 and 1 (probabilities)).[18 20]

## Costs

All cost parameters are standardised to 2019 euros, and summarised in (Online supplemental table S2). Event-related healthcare costs were based on a previous Dutch cost-effectiveness study for rivaroxaban in the general VTE population.[22] Costs of fatal recurrent VTE were assumed to be similar to those of non-fatal symptomatic PE. We assumed no event-related healthcare costs for patients with incidental PE because these embolisms were found incidentally and did therefore not require physician visits. However, since patients with incidental PE should be treated identically to those with symptomatic PE, we did take medication costs into account. Costs for ICH and CTEPH consisted of acute care costs during the first month after diagnosis, followed by long-term care costs until the patient moved to the 'death' state. Costs of a fatal MB were assumed to be equal to those of non-fatal non-ICH MB.

Drug costs were retrieved from the national medication costs database.[23] For rivaroxaban, these costs were based on 15 mg two times per day for 3 weeks followed by 20 mg once daily. Drug costs of dalteparin were based on 200 IU/kg daily during month 1 followed by 150 IU/kg daily in months 2–6.[15 24] Based on an average body mass index of 25.6 from the SELECT-D trial and an average height of 1.72 m for the Dutch population, we calculated that the average weight was between 69 and 82 kg, which corresponds with a dose of 15 000 IU daily during month 1 followed by 12 500 IU daily in months 2–6.[15 25] Rivaroxaban users were assumed to require an annual check-up of their renal function.[6] We included one-time costs for an injection instruction by a home caregiver. Administration costs were only accounted to patients with early or locally advanced cancer (39%), since patients with metastatic cancer or haematologic malignancies often already have home care or an informal caregiver who can administer the dalteparin injection. Similarly, informal care costs were only taken into account for this same subgroup.

Based on a previously published report on informal care in the Netherlands, we made a distinction between intensive (26 hours per week) and non-intensive (8 hours per week) informal care.[26] This was multiplied by the average duration and tariff for informal care, obtained from the Dutch cost manual.[27] To prevent double counting, we did not include informal care costs for the chronic complications. Travel costs were taken into account for renal monitoring visits and on the occurrence of a DVT or CRNMB. Costs related to forgone leisure activity were not taken into account since there are no data available on the impact of a VTE or bleeding on leisure losses in patients with cancer. Moreover, the starting age of the population in the model was 67 years (which is the Dutch retirement age) based on the average age of the SELECT-D trial and the fact that the majority (58%) of the patients in the SELECT-D trial had metastatic cancer may indicate a low employment rate.

Costs were discounted at an annual rate of 4%.[18] In the sensitivity analysis, the costs were varied with gamma distributions corresponding to the 95% CI,[18 20] as indicated in online supplemental table S2.

## Utilities

Utility scores, used to calculate the QALYs, were derived from a subanalysis from the CATCH study assessing the EQ-5D scores associated with VTE and recurrent VTE in patients with cancer (online supplemental table S3).[28] The CATCH study assessed the effectiveness of 6 months of treatment with tinzaparin versus warfarin for the treatment of acute VTE in patients with active cancer. It was chosen because it aligns well with our population and events of interest. Utility decrements for CTEPH were based on a study assessing EuroQol-5D-visual analogue scale (EQ-5D VAS) scores in CTEPH patients up to 5 years after their initial diagnosis.[29] Utility decrements

**Table 2** Overview of the scenario analyses

| Scenario | Description | Details |
|---|---|---|
| Base case | 5-year time horizon from societal perspective | – |
| 1 | 6-month time horizon from societal perspective | The follow-up period of the SELECT-D trial was 6 months; therefore, outcomes beyond 6 months had to be extrapolated based on other publications. |
| 2 | Base case analysis from healthcare payer's perspective | In the Netherlands, guidelines advise to calculate the ICER from a societal perspective, while in countries such as the UK or Belgium, the healthcare payer's perspective is preferred. To make results comparable with other countries, we also calculated the base case ICER from a healthcare payer's perspective, by excluding the indirect costs. |
| 3<br><br>4 | Base case analysis with dalteparin dose of 12 500 IU<br><br>Base case analysis with dalteparin dose of 18 000 IU | The costs of dalteparin vary with the patient's weight. For the base case analysis, we assumed an average weight between 69 and 82 kg. In scenarios 3 and 4, we calculated the base case ICER with the costs of dalteparin based on weight categories of 57–68 kg (12 500 IE daily during month 1 followed by 10 000 IE daily in months 2–6) and 83–98 kg (18 000 IE daily during month one followed by 15 000 IE daily in months 2–6), respectively. |
| 5 | Scenario one with treatment duration based on Streiff et al[17] | This scenario was similar to scenario 1, except for the treatment period which was based on a study of Streiff et al, who—comparable to SELECT-D—compared rivaroxaban to LMWH for the prevention of recurrent VTE in cancer patients.[17] They found an average treatment duration of 1 and 3 months for LMWH and rivaroxaban, respectively. |
| 6 | Base case analysis using drug-specific distributions for the types of VTE and MB | Due to low numbers of VTE and MB events observed in the SELECT-D trial[15] and HOKUSAI VTE Cancer trials, respectively, we calculated the distribution of the types of VTE and MB in the base case analysis based on the total number of events and assumed it to be equal for both drugs. In this scenario, we assess the effect of this assumption on the cost-effectiveness results by using the drug-specific distributions of the types of VTE and MB based on the results of the SELECT-D and HOKUSAI VTE Cancer trials.[15 16] |

HOKUSAI VTE, Edoxaban for the Treatment of Cancer-Associated Venous Thromboembolism; ICER, incremental cost-effectiveness ratio; IU, international units; LMWH, low molecular weight heparin; MB, major bleeding; SELECT-D, Comparison of an Oral Factor Xa Inhibitor With Low Molecular Weight Heparin in Patients With Cancer With Venous Thromboembolism: Results of a Randomized Trial; VTE, venous thromboembolism.

for ICH and long-term PTS (>6 months after diagnosis) were obtained from a previous cost-effectiveness study.[30] QALYs related to fatal events, such as death due to any cause, fatal PE and fatal MB, were assumed to be 0. QALYs were discounted at 1.5% per annum according to Dutch guidelines.[18] In the sensitivity analyses, utility scores were varied over their 95% CI with a beta distribution.[18 20]

### Sensitivity analysis

Sensitivity analyses were conducted to check the robustness of the model results to uncertainty and known variations in key input parameters. In the probabilistic sensitivity analysis, all input parameters were varied simultaneously over their 95% CI. If the 95% CI was unavailable and calculating the 95% CI based on the number of events was not possible, the 95% CI was calculated based on a 25% SE. The ICER was calculated with 2000 iterations and plotted in a cost-effectiveness plane. A univariate sensitivity analysis was conducted to show the influence of an individual parameter on the ICER. The 10 most influencing parameters were presented in a tornado diagram.

### Scenario analysis

We conducted several scenario analyses to show the effect on the outcomes of different (clinical) situations (table 2).

### Budget impact

A budget impact analysis was conducted to estimate the total annual financial consequences of the implementation of rivaroxaban for the treatment and prevention of VTE in patients with cancer within the Dutch healthcare setting. The budget impact was calculated from a societal perspective using the costs calculations from the cost-effectiveness model with a 1-year time horizon. We extracted from the model the costs (event-related, treatment and indirect costs) per patient with a cut-off point of 1 year for rivaroxaban and dalteparin. The difference in cost per patient was multiplied by the annual number of cancer patients with VTE in the Netherlands. The incidence of VTE in patients with cancer and the total number of Dutch cancer patients were used to calculate the yearly number of cancer patients with VTE. The Netherlands Cancer Registry estimated a total of 579 781 patients with cancer in 2017.[31] The incidence of VTE in patients with cancer was 13.9 per 1000 person-years,

**Table 3** Deterministic results per patient of the base case and scenario analyses in a cohort of 1000 patients with cancer (2019, euros)

| | Costs | QALYs | ΔCosts | ΔQALYs | ICER |
|---|---|---|---|---|---|
| Base case analysis: 5-year time horizon from societal perspective | | | | | |
| Rivaroxaban | €3139 | 2.459 | −€1476 | 0.012 | Rivaroxaban dominant |
| Dalteparin | €4615 | 2.448 | | | |
| Scenario 1: 6-month time horizon from societal perspective | | | | | |
| Rivaroxaban | €1361 | 0.304 | −€1312 | 0.004 | Rivaroxaban dominant |
| Dalteparin | €2673 | 0.300 | | | |
| Scenario 2: base case analysis from healthcare payer's perspective | | | | | |
| Rivaroxaban | €2942 | 2.459 | −€1496 | 0.012 | Rivaroxaban dominant |
| Dalteparin | €4438 | 2.448 | | | |
| Scenario 3: base case analysis with dalteparin dose of 12 500 IU | | | | | |
| Rivaroxaban | €3139 | 2.459 | −€1079 | 0.012 | Rivaroxaban dominant |
| Dalteparin | €4218 | 2.448 | | | |
| Scenario 4: base case analysis with dalteparin dose of 18 000 IU | | | | | |
| Rivaroxaban | €3139 | 2.459 | −€1898 | 0.012 | Rivaroxaban dominant |
| Dalteparin | €5037 | 2.448 | | | |
| Scenario 5: scenario one with treatment duration based on Streiff et al[17] | | | | | |
| Rivaroxaban | €1299 | 0.289 | −€702 | 0.016 | Rivaroxaban dominant |
| Dalteparin | €2001 | 0.273 | | | |
| Scenario 6: base case analysis using drug-specific distributions for the types of VTE and MB | | | | | |
| Rivaroxaban | €3065 | 2.463 | −€1815 | 0.037 | Rivaroxaban dominant |
| Dalteparin | €4880 | 2.425 | | | |

ICER, incremental cost-effectiveness ratio; IU, international units; MB, major bleeding; QALY, quality adjusted life-years; VTE, venous thromboembolism.

based on a cohort study of linked UK databases.[32] Based on these numbers, we calculated a total of approximately 8000 cancer patients with VTE per year in the Netherlands. The outcome of the budget impact analysis was presented as the total budget impact per year, including a subdivision of the costs per type (event-related costs, treatment costs and indirect costs) and corresponding 95% CIs derived from PSA.

### Patient and public involvement

It was not appropriate to involve patients or the public in the design, or conduct, or reporting or dissemination plans of our research, because this health economic analysis was based on publicly available data and solely concentrated on the analysis of the economics consequence of treating cancer patients with rivaroxaban instead of the current standard of care.

### RESULTS
### Cost-effectiveness analysis

Table 3 represents the deterministic results of the base case and scenario analyses. In each scenario, rivaroxaban was economically dominant—meaning that it simultaneously confers better clinical and quality-of-life outcomes at less cost—compared with dalteparin.

As such, a numerical ICER is not presented because it has no meaning. Despite the fact that every scenario shows an improvement in the patient's health, the difference in QALYs was very low (incremental QALYs of 0.012 over 5 years' time horizon, which equals 4.4 quality-adjusted life days, in the base case analysis). In the base case analysis, rivaroxaban saved €1376 per patient compared with dalteparin. The scenario calculating the cost-effectiveness over a 6-month time horizon resulted in cost-savings of €1312 per patient (scenario 1). There was increased cost-savings compared with the societal perspective when calculated from a healthcare payer's perspective (scenario 2). In scenarios 3 and 4, we assessed the effect of variations in the patient's weight (and thus dalteparin dosing) on the ICER. Compared with the base case analysis, there was decreased cost-savings with a lower dalteparin dose and increased cost-savings with a higher dalteparin dose, both still resulting in dominant ICERs. When comparing 3 months of rivaroxaban treatment to 1 month of dalteparin treatment, we found incremental QALYs of 0.016 and cost-savings of €702 per patient (scenario 5). We assessed the effect of using drug-specific distributions of the types of VTE and MB, resulting in cost-savings of €1815 and incremental QALYs of 0.037 (scenario 6).

**Table 4** Number of events and costs per event per patient in a cohort of 1000 patients with cancer (2019, euros)

| | Rivaroxaban | | Dalteparin | | Incremental | |
|---|---|---|---|---|---|---|
| | No of events | Costs per patient | No of events | Costs per patient | No of events | Costs per patient |
| **Base case (5-year time horizon)** | | | | | | |
| Event costs | | | | | | |
| Recurrent VTE | 191 | €311.85 | 275 | €442.92 | −84 | −€131 |
| Non-fatal symptomatic recurrent PE | 33 | €168.36 | 48 | €239.13 | −15 | −€71 |
| Non-fatal incidental recurrent PE | 58 | – | 84 | – | −26 | |
| Non-fatal recurrent DVT | 83 | €59.31 | 120 | €84.23 | −37 | −€25 |
| Fatal recurrent VTE | 17 | €84.18 | 24 | €119.56 | −7 | −€35 |
| ICH | 11 | €550.70 | 9 | €438.40 | 2 | €112 |
| Non-ICH MB | 98 | €1106.87 | 79 | €902.47 | 19 | €204 |
| Fatal MB | 5 | €51.48 | 4 | €41.98 | 1 | €10 |
| CRNMB | 197 | €56.28 | 92 | €26.93 | 105 | €29 |
| PTS | 61 | €92.72 | 61 | €92.37 | 0 | €0 |
| CTEPH | 20 | €223.79 | 20 | €222.83 | 0 | €1 |
| Total event costs | | €2705.54 | | €2610.83 | | €95 |
| Treatment costs | | €548.83 | | €2270.33 | | −€1721 |
| Indirect costs | | €196.31 | | €177.08 | | €19 |
| **Scenario 1 (6-month time horizon)** | | | | | | |
| Event costs | | | | | | |
| Recurrent VTE | 38 | €58.95 | 109 | €166.96 | −70 | −€108 |
| Non-fatal symptomatic recurrent PE | 7 | €31.82 | 19 | €90.14 | −12 | −€58 |
| Non-fatal incidental recurrent PE | 12 | – | 33 | – | −21 | – |
| Non-fatal recurrent DVT | 17 | €11.21 | 47 | €31.75 | −31 | −€21 |
| Fatal recurrent VTE | 3 | €15.91 | 9 | €45.07 | -6 | −€29 |
| ICH | 6 | €142.82 | 4 | €94.25 | 2 | €49 |
| Non-ICH MB | 50 | €539.38 | 33 | €355.95 | 17 | €183 |
| Fatal MB | 2 | €25.09 | 2 | €16.56 | 1 | €9 |
| CRNMB | 130 | €35.99 | 38 | €10.62 | 91 | €25 |
| PTS | 14 | €20.59 | 14 | €20.56 | 0 | €0 |
| CTEPH | 3 | €21.96 | 3 | €21.93 | 0 | €0 |
| Total event costs | | €903.72 | | €2639.25 | | −€1736 |
| Treatment costs | | €479.40 | | €1947.45 | | −€1468 |
| Indirect costs | | €36.50 | | €38.39 | | −€2 |

CRNMB, clinically relevant non-major bleeding; CTEPH, chronic thromboembolic pulmonary hypertension; DVT, deep vein thrombosis; ICH, intracranial haemorrhage; MB, major bleeding; PE, pulmonary embolism; PTS, post-thrombotic syndrome; VTE, venous thromboembolism.

The number of events and the corresponding average costs per patient in the base case analysis and scenario 4 (base case analysis with a time horizon of 6 months) are presented in table 4. Rivaroxaban is associated with a lower number of recurrent VTE events, preventing on average €131 and €108 in costs per patient over 5 years and over 6 months, respectively. On the other hand, rivaroxaban causes more bleeding events, especially in the treatment period. ICH and non-ICH MB have the highest incremental event costs per patient. Treatment costs are higher for dalteparin compared with rivaroxaban, with incremental costs of €1721 and €1468 in the 5-year and the 6-month time horizon, respectively. The differences in indirect costs for rivaroxaban compared with dalteparin were €19 and −€2 for the 5-year and the 6-month time horizon, respectively.

In the probabilistic sensitivity analysis, we assessed the robustness of the model over a 5-year time horizon (base case) and a 6-month time horizon (scenario 1). The results are presented in cost-effectiveness planes in

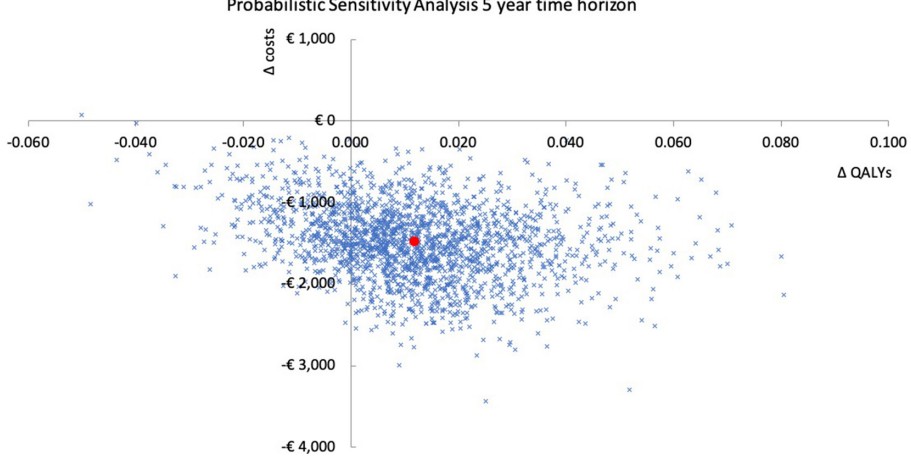

**Figure 2** Probabilistic sensitivity analysis of the base case with 5-year time horizon (base case analysis). The red mark represents the deterministic incremental cost-effectiveness ratio. QALY, quality adjusted life-year.

figure 2 and online supplemental figure S1. In the base case analysis, rivaroxaban was in the majority (77.8%) of the 2000 iterations cost-saving and more effective compared with dalteparin. In 22.2% of the iterations, rivaroxaban was cost-saving but less effective compared with dalteparin. In scenario 1, rivaroxaban was in almost all (98.7%) the iterations cost-saving and more effective compared with dalteparin.

The influence of the individual input parameters on the base case incremental costs and QALYs is analysed in the univariate sensitivity analysis. The tornado diagrams (figures 3 and 4) present the 10 input parameters with the highest impact in the base case analysis. The risk of MB for both rivaroxaban and dalteparin, treatment duration of dalteparin and recurrent VTE risks during the first 6 months after a VTE had the highest influence on the

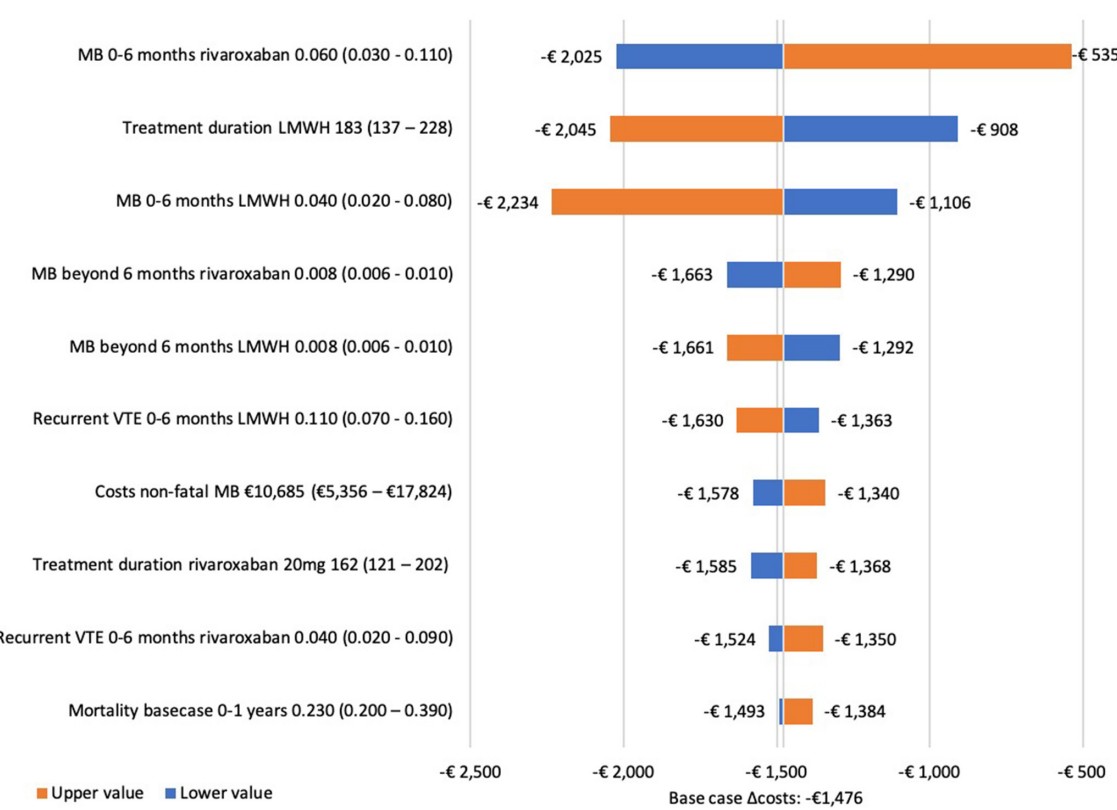

**Figure 3** Tornado diagram from the univariate sensitivity analysis for the base case analysis showing the impact of parameters on the incremental costs. ICH, intracranial haemorrhage; LMWH, low molecular weight heparin; MB, major bleeding; PE, pulmonary embolism; VTE, venous thromboembolism.

de Jong LA, et al. BMJ Open 2020;**10**:e039057. doi:10.1136/bmjopen-2020-039057

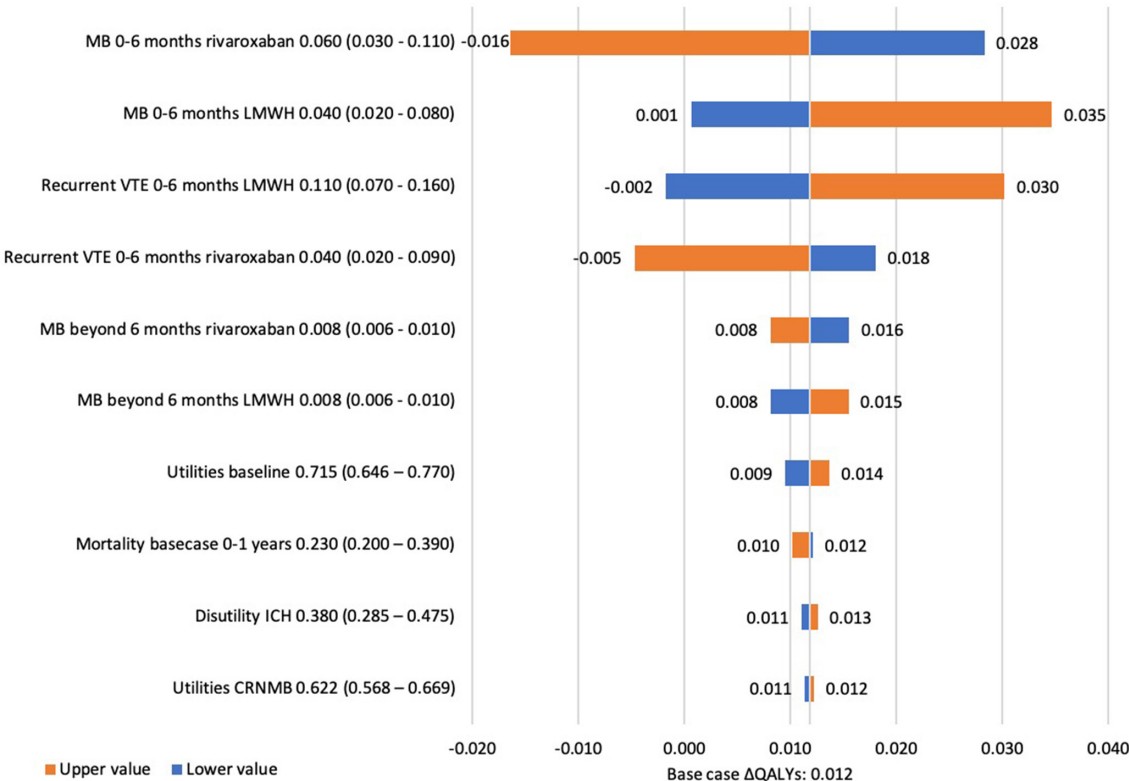

**Figure 4** Tornado diagram from the univariate sensitivity analysis for the base case analysis showing the impact of parameters on the incremental QALYs. CRNMB, clinically relevant non-major bleeding; ICH, intracranial haemorrhage; LMWH, low molecular weight heparin; MB, major bleeding; VTE, venous thromboembolism; QALY, quality adjusted life-year.

incremental costs. Similarly, the risk of MB and recurrent VTE in the first 6 months for rivaroxaban and dalteparin showed the highest influence on the incremental QALYs. Similar results were found in the univariate sensitivity analysis of scenario 1 (online supplemental figure S2 and figure S3).

### Budget impact

The results of the budget impact analysis are presented in table 5. The replacement of LMWHs (including dalteparin) with rivaroxaban can lead to cost-savings of a maximum of €11 326 763 (€5 164 254–€17 363 231) over approximately 8000 cancer patients with VTE based on a 1-year time horizon. A reduction in treatment costs can lead to savings of up to €12.6 million. Event-related costs and indirect costs slightly increase by €1 234 467 (–€2 103 366 to €5 231 955) and €2101 (–€173 830 to €184 677), respectively, when LMWHs are replaced by rivaroxaban.

### DISCUSSION

Thrombosis treatment is a challenge in patients with cancer. According to the guidelines, LMWHs and DOACs edoxaban and rivaroxaban are the preferred treatment for the prevention of recurrent VTE in patients with cancer.[8–11] We have assessed the cost-effectiveness and budget impact of rivaroxaban in patients with cancer at risk of recurrent VTE based on the SELECT-D trial.[15] We conclude that, in the Netherlands, rivaroxaban is a cost-saving treatment option with a small health benefit per patient over 5 years compared with dalteparin. Comprehensive sensitivity analyses confirm that results generated by our model are robust.

The cost-savings associated with rivaroxaban were mainly driven by the difference in treatment costs. It should be noted that this is specifically the case for the Netherlands and may differ in other countries. The VTE recurrence and MB risks also showed to have a high influence on the incremental costs and QALYs in the univariate sensitivity analysis. The SELECT-D trial showed a relatively low VTE recurrence but higher bleeding (especially CRNMB) compared with dalteparin. This

**Table 5** Budget impact (95% CI) over 1-year time horizon in the Netherlands

| Event-related costs | €1 234 467 (–€2 103 366 to €5 231 955) |
| Treatment costs | –€12 559 130 (–€17 327 405 to –€8 149 498) |
| Indirect costs | –€2101 (–€173 830 to €184 677) |
| Budget impact | –€11 326 763 (–€17 363 231 to –€5 164 254) |

cost-effectiveness model allowed to address the question if the reduction in VTE recurrence outweighs the increase in bleeding events.

A total of 84 VTE-related events were prevented over 5 years, leading to an average cost-saving of €131 per patient. This is line with findings from a recent study that assessed the VTE-related healthcare costs in patients with cancer, which found that rivaroxaban-treated patients had a significantly lower total VTE-related costs relative to patients treated with LMWH.[33] Although the cost difference between the rivaroxaban and dalteparin cohorts was even greater with $12 004 per patient per year.

On the other hand, MB events were more frequent with rivaroxaban compared with dalteparin (11 ICH and 98 non-ICH vs 9 ICH and 79 non-ICH, respectively). MB events are very burdensome and frequently severely disabling, leading to high acute and long-term direct and indirect costs. In line with the findings from the SELECT-D trial, CRNMB events were much more frequent with rivaroxaban compared with dalteparin (197 and 92, respectively). Although the difference between rivaroxaban and dalteparin in CRNMB (105 events over 5-year time horizon) is greater than for MB (20 events over 5-year time horizon), the influence on the incremental costs and QALYs was lower because CRNMB is relatively less burdensome.

The indirect costs were higher for rivaroxaban than for dalteparin in the base case scenario. This can be explained by the increased number of MB events with rivaroxaban compared with dalteparin. Moreover, there were no data available on leisure activity losses caused by the occurrence of a VTE event in patients who are already burdened with cancer. Therefore, the indirect costs might have been underestimated, possibly leading to lower cost-savings. The indirect costs account for €196–€177 per rivaroxaban and dalteparin patient, respectively, over 5 years—approximately 4%–6% of the total cost—however, they do not have a major influence on the differences between the two drugs (€19 and –€2 for the 5-year and 6-month time horizon, respectively). This suggests that, although the indirect costs might have been underestimated, rivaroxaban is still likely to be cost-saving compared with dalteparin.

As mentioned, the main driver of the cost-savings is the difference in treatment costs. In the cost-effectiveness analysis, we estimated that more than €1700 per patient over a 5-year period can be saved on treatment costs, compared with dalteparin. Moreover, in the scenario analysis, we varied the price of dalteparin based on weight. Although the lowest dose (12 500 IU daily during month 1 followed by 10 000 IU in months 2–6 based on weight class 57–68 kg) had a lower price, €8.06 vs €9.93, the ICER remained cost-saving. Rivaroxaban users were assumed to require an annual check-up of their renal function. However, patients with cancer (especially those with metastatic cancer) are at higher risk for renal impairment and may be tested much more frequently.[34] This may have caused an overestimation of the costs of rivaroxaban, and

therefore reduced the cost-savings estimate of rivaroxaban compared with dalteparin.

In the budget impact analysis, we calculated that rivaroxaban replacing LMWH (including dalteparin) leads to cost-savings of €11 326 763 within 1 year over a total of 8000 patients with cancer. This is the absolute maximum, since it is not possible to treat each patient with rivaroxaban from a clinical perspective. In practice, the market share of rivaroxaban will be lower—despite the fact that there are three other DOACs that could be prescribed—because there are some clinical considerations that should be taken into account. First, although DOACs have far fewer drug interactions than VKAs, it should be noted that rivaroxaban is metabolised by CYP3A4 enzymes.[1] Patients with cancer, especially those with haematological cancer, are at high risk for opportunistic and fungal infections, for which they are often treated with CYP3A4 inhibitors or inducers.[35] For this reason, prescription of rivaroxaban for the prevention of recurrent VTE in patients with cancer must be done carefully.[1] This interaction does not play a role in LWMH treatment.

Second, the balance between the risk of thrombosis and the risk of bleeding should always be a consideration in the prescription of anticoagulants. For example, DOACs are not advised in patients with gastrointestinal (GI) tumours, due to a higher risk of GI bleeding.[8–11] Some prediction scores for primary prevention have been developed to predict thrombosis risk in patients with cancer, since thrombosis prophylaxis is most effective in patients with an increased VTE risk. Unfortunately, for cancer these scores have still not been shown to reliably identify patients with the highest risk.[36] Predictive scores for bleeding, such as the HAS-BLED score used for atrial fibrillation patients, are also needed.

A third consideration is the oral administration of rivaroxaban. Although it is less burdensome than the LMWH injections, oral administration can be problematic in patients with anorexia and vomiting, which is often seen as a side effect in cancer therapy.[15] Moreover, low food intake might influence the metabolism of rivaroxaban resulting in lower bioavailability.[37] Lastly, adherence is always a point of discussion, but since adherence to current guidelines is often low,[36] we feel that adherence to rivaroxaban might be relatively high than LMWHs due to the more patient-friendly administration.

Our analysis is not without limitations. It should also be noted that 58% of the patients included in the SELECT-D trial had metastatic cancer, and thus results and conclusions pertain mostly to severely ill patients. Also, the majority (53%) of the initial VTE events were incidental PE, related to CT imaging for tumour status.[15] Additionally, as with all cost-effectiveness models some assumptions were required due to lack of data.

We assumed that patients were treated with anticoagulation over 6 months, which is in line with the guidelines.[8–11] Previous studies have shown that adherence to these guidelines is poor.[36] As seen in the study by Streiff et al,[17] in practice, treatment with LMWH is often not 6 months,

presumably due to the fact that LMWH injections are burdensome, there are concerns about the bleeding risk, and the complexity of the treatment of patients with cancer.[36] However, this recommended treatment period was also not achieved in many patients treated with rivaroxaban, which resulted in an average duration of 3 months. We conducted a scenario analysis (scenario 5) to assess this difference in treatment duration (1 month of LMWH vs 3 months of rivaroxaban). These results favoured rivaroxaban because the incremental QALYs increased while still being cost-saving. On the other hand, there are also some clinical situations in which the treatment period might be longer than 6 months: for example, in patients with a recurrent VTE event, patients with an active malignancy, or patients receiving cancer treatment for their malignancy beyond 6 months. Moreover, in the Netherlands anticoagulation is often continued after 6 months of initial treatment in case the cancer is still active. Unfortunately, we were unable to assess the effect of continued anticoagulation treatment due to lack of data. However, since rivaroxaban is associated with cost-saving results during the first 6 months, it is to be expected that during a longer treatment period the cost-savings and health gains will accrue even more compared with dalteparin.

In the univariate sensitivity analysis, we have shown that the risks of MB and VTE for both rivaroxaban and dalteparin have a high influence on the incremental costs and QALYs. In the SELECT-D trial,[15] the incidence of symptomatic and fatal PE events was relatively higher in patients treated with rivaroxaban. However, due to low numbers of VTE observed in the SELECT-D trial,[15] we calculated the distribution of the type of VTE based on the total number of events and assumed it to be equal for both drugs. This may have led to an overestimation of the effect of rivaroxaban compared with dalteparin, since symptomatic and fatal PE events have a higher impact on the costs and the patient's health compared with DVT and incidental PE. On the other hand, we used this same approach to calculate the distributions of the types of MB from the HOKUSAI VTE Cancer trial,[16] in which the patients treated with dalteparin had relatively more severe MB events compared with the NOAC edoxaban (ICH: 17.6% vs 6.1%, respectively). This results in an underestimation of the number of MBs in dalteparin-treated patients. We assessed the effect of using drug-specific distributions of the type of VTE and MB in scenario 6, showing an increase in incremental cost-savings and QALYs compared with the base case analysis. Therefore, we conclude that our approach of using equal distributions of the types of VTE and MB for rivaroxaban and dalteparin is conservative.

This study focuses on the secondary prevention of VTE, based on the results of the SELECT-D and, partially, the HOKUSAI VTE Cancer trials. However, recently, apixaban was also assessed in patients with cancer at risk of recurrent VTE and found to be non-inferior compared with dalteparin.[38 39] Moreover, the AVERT and CASSINI trials have shown that apixaban and rivaroxaban are also effective as a primary prophylaxis of VTE in patients with cancer compared with a placebo.[40–42] Based on these two studies, clinicians may consider DOAC prophylaxis in some of their patients with cancer.[42] Therefore, future research is needed to assess if DOACs are also cost-effective for the primary prevention of VTE.

## CONCLUSION

Treatment with rivaroxaban is dominant (cost-saving while slightly improving the patient's health and quality of life) over dalteparin in patients with cancer at risk for recurrent VTE in the Netherlands. The use of rivaroxaban instead of LMWH (including dalteparin) can save more than €11 million per year, which is primarily driven by the difference in treatment costs. Since treatment with rivaroxaban is economically dominant compared with dalteparin and its oral administration is more convenient than daily subcutaneous injection, it is logical that certain patients with cancer can benefit from DOAC treatment and provide savings to the healthcare system.

**Author affiliations**
[1]Unit of Pharmacotherapy, Epidemiology and Economics, University of Groningen, Groningen, The Netherlands
[2]Department of Internal Medicine, Martini Hospital, Groningen, The Netherlands
[3]Department of Clinical Pharmacy and Toxicology, Martini Hospital, Groningen, The Netherlands
[4]Department of Health Sciences, University Medical Centre Groningen, Groningen, The Netherlands
[5]Department of Economics, Econometrics & Finance, University of Groningen, Groningen, The Netherlands

**Contributors** LAdJ built the economic model, performed the analyses and contributed to the design of the work, interpretation of the results, writing of the manuscript. AWGvdV contributed to the interpretation of the results, writing of the manuscript and critical revision for important intellectual content. MvH contributed to the design of the work, interpretation of the results, writing of the manuscript and critical revision for important intellectual content. MJP contributed to the design of the work, interpretation of the results, writing of the manuscript and critical revision for important intellectual content. All authors approved of the version to be published.

**Funding** This work was supported by Bayer Pharma Netherlands. The sponsor was involved with the start of the project, but they were not involved in the identification of data, design, conduct and reporting of the analysis. Award/grant number: not applicable.

**Competing interests** LAdJ, MvH and AWGvdV declare that they have no competing interest with relation to subject. MJP has received research grants from various pharmaceutical companies, including but not limiting to Bayer, Pfizer, Bristol-Myers Squibb, GSK, Roche and Novartis.

**Patient consent for publication** Not applicable.

**Ethics approval** The analyses were conducted based on publicly available information which is presented and referenced in the article and Supporting Information files, and did therefore not require any patient consent forms or approval from an ethical review board.

**Provenance and peer review** Not commissioned; externally peer reviewed.

**Data availability statement** All relevant data are included in the manuscript. The analyses were conducted based on publicly available information which is presented and referenced in the manuscript.

responsibility arising from any reliance placed on the content. Where the content includes any translated material, BMJ does not warrant the accuracy and reliability of the translations (including but not limited to local regulations, clinical guidelines, terminology, drug names and drug dosages), and is not responsible for any error and/or omissions arising from translation and adaptation or otherwise.

**ORCID iD**
Lisa A de Jong http://orcid.org/0000-0001-8814-0670

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
