## [Reviewer comments · BMJ Open]

ARTICLE DETAILS

TITLE (PROVISIONAL)	Cost-effectiveness analysis and budget impact of rivaroxaban compared with dalteparin in cancer patients at risk of recurrent venous thromboembolism
AUTHORS	de Jong, Lisa; van der Velden, Annette; Hulst, Marinus; Postma, Maarten

VERSION 1 – REVIEW

REVIEWER	Annie Young The University of Warwick, United Kingdom Competing Interest: I am the principal investigator for the study on which this analysis was initially based.
REVIEW RETURNED	07-May-2020

GENERAL COMMENTS	This is a health economic modelling paper, exploring cost effectiveness and budget impact of rivaroxaban and low molecular weight heparin in the cancer population at risk of recurrent VTE, hitherto not carried out using this approach, although this paper needs to be put into the context of recent publication from Strieff et al, J Med Econ. 2019 Nov;22(11):1134-1140, to be strengthen the case. General Comments: I am a clinician and therefore commenting on the clinical aspects of the paper. In this respect, I feel it would be beneficial to have further input from clinical experts on cancer and thrombosis – mainly to update/correct some of the text and include more recent references. Bayer, the makers of rivaroxaban, funded the study and there may be a potential conflict; I do not see a statement on the role of the funder in the identification, design etc of the analysis as in Item 23 of CHEERS checklist, which would indicate no conflict or conflict. The costs assumptions look good and are applicable to the Netherlands and the scenarios. I welcomed scenario 5 in particular as treatment duration of 6 months does not generally reflect practice in this population, as the authors discuss. The treatment costs which drive the cost benefit analysis may differ in other countries; although implicit in the paper, this should be highlighted. Specific Comments: 1. Abstract and conclusion: ‘...many cancer patients can benefit...’ I would tone that down to ‘certain cancer patients’, due to clinical challenges which the authors address in the discussion
--

	2. References for Guidelines on treatment and prevention of recurrence of VTE in the cancer population are out of date (refs 4 and 7-9) so more recent guidelines e.g. ITAC, ASCO update, NCCN update, have different wording in their recommendations as stated in this paper. There appears to be differences, between the introduction and the discussion, in the summary of what the international guidelines state. Lines 96 and 97 are wrong. 3. Line 101 – “Patients with UGI cancers were excluded” – this is inaccurate as these patients were not excluded until nearing the end of the study (ref 44). And select-d did include all patients with GI tumours, in fact the biggest group was patients with colorectal cancer (line 409; reference 14). 4. I appreciate that the model may have been started many months ago; however the 12 month data for the select-d trial I believe were published online in January 2020 and may have provided continuity for the 12 month current state (lines 256, 179 and 435). It would be helpful to state when the analysis was carried out. 5. Discuss the meaning of the estimated health gain of 0.012 QALY with rivaroxaban relative to LMWH e.g. does this fall above or below a threshold for minimally important differences seen in other studies in similar settings? Minor comment: The word data is plural. In conclusion, the model and assumptions made seem reasonable; the health economists have carried out a thorough analysis. However, the clinical rationale for the health economic analyses needs updated and amended. The difference in the budget impact in favour of rivaroxaban is relevant for the Netherlands.
--	--

REVIEWER	Jeffrey D. Miller IBM Watson Health Cambridge, Massachusetts, USA
REVIEW RETURNED	19-May-2020

GENERAL COMMENTS	Please see my detailed comments with line references in the attached file. This is a well-written paper with solid underpinnings of data and methods. All the criteria for good modeling practices were met or exceeded. I prepared a long list of comments and suggested edits which I hope the authors do not find too daunting. Most of the criticisms are relatively minor, and I do not think new analyses or substantial re-writing needs to be undertaken. There was a flawed assumption about "negligible" productivity losses because average patient age in the model is 67 years, but this was unlikely to have any major impact on the analysis results. Some minor re-work should quickly fix this. I do suggest that the authors focus on the fact that this specifically is a rivaroxaban vs. dalteparin analysis, and not try to step out those bounds into portraying this as a DOAC vs. LMWH class analysis. Further, this is an analysis of secondary prevention or prophylaxis of recurrent VTE, not treatment of VTE. The authors are a bit inconsistent with their terms and unintentionally entangle "prevention" and "treatment," but only in their choice of words. Nonetheless, care should be taken to make the distinction. I think some editorial polish will set these and most other things straight.
--

	I extend my congratulations to the authors for a study well done. Reviewer comments for de Jong et al. (Cost-Effectiveness Analysis and Budget Impact of Rivaroxaban in Cancer Patients at Risk of Recurrent Venous Thromboembolism) Title: State openly in the title that this is a “rivaroxaban vs. dalteparin” analysis. Otherwise the reader is led to believe that the analysis is a broad comparison of rivaroxaban against all competing therapies, which is not the scope of this study. Also, make this clear in the text too, as sometimes “dalteparin” seems hidden behind the nomenclature of “LMWH.” It is appropriate to mention once that dalteparin is a LMWH, but thereafter the comparator should be distinctly termed “dalteparin.” Line 31: State that the comparison is with dalteparin, not the broad class of LMWH. As mentioned above, it is appropriate to mention once that dalteparin is a LMWH, but thereafter the comparator should be distinctly termed “dalteparin.” If you talk about dalteparin in context of LMWHs, then you need to talk about rivaroxaban in context of DOACs. Line 43: State who or what organization bears the “financial consequences” Line 44: Replace “appeared to be” with “were” – your analysis findings are observationally concrete and not interpretive (also see Line 283, where this same comment applies) Line 45: Replace “increasing” with “improving” Line 45: “Dominant” is usually termed “economically dominant” (also Line 49) Line 46: Replace “were cost saving” with “showed rivaroxaban to be cost saving” Line 47: The “€9,834,144” is an annual average, given that you are looking at time horizons both longer and shorter than one year. Line 52: Mention of “less invasive” is strange here; perhaps another word choice would be appropriate Lines 52-53: The conclusion falls flat, as the authors step out of the context of the study, which is a financial analysis, and into the vague area of patient benefit. Isn’t the healthcare payer the focus stakeholder in this study? Lines 52-53: This is the first place you mention “direct oral anticoagulant”; you should state earlier that rivaroxaban is in the class of DOAC anticoagulants. Line 59: Probably should add “Markov” to “tunnel states” – i.e., “Markov tunnel states” Line 65: Seems odd to mention apixaban here Line 69: Be more specific about the Bayer Pharma funding – global or a local affiliate? Line 82: Replace “prophylaxis” with “prophylaxis for recurrence” Line 87: As mentioned above, you are singling out dalteparin without much explanation about the rationale Line 93: Provide examples of DOACs. Isn’t this where you want to introduce rivaroxaban as a DOAC?
--	--

	Line 96: Although they may be “not recommended” aren’t DOACs commonly used in real-world practice? You might want to mention this. Line 100: It comes later on Lines 132-133, but suggestion to move the explanation of “incidental PE” to here – many readers will not know Line 101: Replace “they” with “the trial researchers” Line 107: Replace “an increase in the use” with “greater utilization” Line 111: Use your acronym VTE for “venous thromboembolism” Line 115: Replace “LMWH” with “dalteparin” (see comments above about how you should always be specific that the comparator is dalteparin) Line 124: The focus of the analysis is on rivaroxaban in a very specific indication; here it sounds like you are broadly evaluating rivaroxaban in all uses Line 129: State what the software platform was. Excel? TreeAge? Line 135: Specifically talk about cancer here and how cancer aligns with the SELECT-D protocol. Otherwise, readers will wonder “what kind of cancer?” Line 137: Break into two sentences Lines 138-139: Need to mention in the Discussion that 58% of patients having metastatic cancer might be a limitation of the analysis. Your results and conclusions pertain mostly to severely ill patients. Also, the majority (53%) of patients have incidental PE – perhaps another limitation? Line 142: Need to state how long “one cycle” is. It comes later on Lines 170-171, but it would benefit to mention here Line 144: State that these are “Markov tunnel states” and explain what that means; typical readers will not know what a tunnel state is and what it does Lines 148-149: Suggest to explain “modeled in the background” – state that incidence and costs of the complications are included in the calculations Line 170: Delete “state per cycle. The cycle length was one month.” and replace with “health state per one-month cycle.” Lines 171-172: Need to reconcile how you have 6-month transition probabilities for 1-month Markov cycles. Probably just need to explain this a bit. Line 173: Replace “the patient” with “patients” Line 179: Replace “this data” with “these data” Lines 181-182: This sentence was difficult to understand and needs to be re-written for better clarity: “The distributions of the types of VTE and MB were calculated based on the total number of events and assumed to be treatment-independent, since the total number of events was low.” Line 183: Remind the reader that you are talking about data abstracted from the SELECT-D trial Line 187: Suggest to explain why “beta distributions” are the best choice – reference to the Briggs paper (#16)? Line 191: Need to state somewhere in this section (perhaps the first sentence) that all costs are standardized to 2019 Euros (as stated in Table S2) Lines 194-195: You state “We assumed no event-related healthcare costs for patients with incidental
--	--

PE.” Why is this? Further explanation is needed.

Lines 197-198: You state “Costs of a fatal MB were assumed to be equal to those of non-fatal 198 non-ICH MB.” But aren’t there substantial indirect costs associated with a fatal event? Family expenses, funeral/burial expenses, estate settlement, etc. Table S2 does not show any indirect costs for fatal events.

Lines 205-206: You state “Rivaroxaban users were assumed to require an annual check-up of their renal function.” Cancer patients (especially those with metastatic disease) are at higher risk for renal impairment and are tested much more frequently. Some investigation into the literature would be revealing here. This omission probably is inconsequential for the model results, but should be checked.

Line 206: What about the cost for a healthcare worker to teach the patient how to self-inject? There is a one-time cost for that. I see in Lines 388-389 you state that you were “conservative” in not including such costs, but it seems more like an omission rather than a safe assumption.

Line 216: You state “productivity losses were assumed to be negligible” because the average patient age was 67 years. However, this logic is flawed because not every patient in the model is exactly 67 years old. Instead, you are essentially modeling patients in a wide distribution of ages, many (but maybe not most) who are of working age and suffer work/productivity impediments. As an illustrative example, the average age of a 55 year old worker and a 79 year old retiree is 67 years.

Line 220: Citation for choice of gamma distribution? The Briggs paper, #16?

Line 224: Mention in this section what the utilities are for – i.e., for calculating QALYs! The reasoning behind this won’t be obvious to many readers.

Line 230 (and Table S3): Unclear how CTEPH beyond 5 years is relevant for this model.

Line 231: Replace “six” with “6”

Line 232: Not correct to say that utilities were discounted, because they were not – it is the QALYs calculated from the utilities that were discounted at 1.5% per annum

Line 237: Change to: “check the robustness of the model results to uncertainty and known variations in key input parameters”

Line 242: You state that there are 15 parameters . . . I see only 12 in the tornado diagrams

Line 246: Suggestion that it might be clearer to the reader to summarize the setup of the scenarios and their assumptions in a table

Line 262: This isn’t clear enough to fully understand: “drug-specific distributions of the types of VTE and MB.”

Line 281: Suggestion to spend a little more time here explaining what is going on with the results with regard to “economic dominance.” You need to explain that that rivaroxaban always simultaneously

	confers better clinical and quality-of-life outcomes at less cost. As such, a numerical ICER is not presented because it has no meaning (hence, why you say “Dominant” in the far-right column of Table 2). This is all perfectly understandable to health economists, but the general readership of the journal is going to be lost without some education about this. Line 284: 0.012 QALYs is equivalent to 4.4 quality-adjusted life days, and that is over 5 years (so about 21 hours per year!). When expressed at this level, it isn’t a particularly impressive figure. In fact, it is almost negligible. As such, rivaroxaban and dalteparin are basically equivalent with regard to promoting quality of life. The authors might want to address this somewhere in the paper, perhaps in the discussion section. Line 287: “scenarios” plural Line 295: Not clear why you look only at Base Case and Scenario 4 here and in Table 3. What about the other scenarios? Lines 298-299: Perhaps you mean that these events have the highest incremental cost or incremental cost differences? Insert “incremental”? Line 310 (Table 2): How come you don’t show a “rolled up” line with the aggregation of event costs? You have lines for Treatment Costs and Indirect Costs, but you are missing the third component of Total Costs. Lines 305-310 (Tables 2 and 3): State the base year for the cost estimates – i.e., “2019 Euros” Lines 318-319: You state “77.0% were located in the south-eastern quadrant,” but that isn’t going to have much meaning to the average reader; suggestion to better explain the meaning and implications Lines 319-320: You state “22.8% are considered cost-saving but less effective compared to LMWH”; given that 1 of every 5 points is in this southwest quadrant, suggestion to better explain the meaning and implications of results that fall into this quadrant, especially how to interpret the nebulous notion of “decremental cost-effectiveness” Line 323: Should you mention in the text that the tornado diagrams are for the Base Case analyses? You do state this in the caption for the diagrams, but probably best to mention in the text too. Also for consideration, you showed probabilistic sensitivity analysis results for Scenario 4, but you chose not to show univariate sensitivity analysis results for Scenario 4? Perhaps show the tornado diagrams for Scenario 4 in the Supplement? Maybe the Scenario 4 scatterplot diagram (Figure 3) should go in the Supplement too, and just show the Base Case results in the main text? Line 324: Again, there are only 12 parameters, not 15 Lines 332 and 337 (Figures 2 and 3): For context, suggestion to put a point of the deterministic result on the plot as a different color (red?); this gives a nice context about how the probabilistic results are arranged around the deterministic result.
--	--

	Lines 353 and 362: Using the available information presented in the tables, I was not able to replicate the calculations that yielded these budget impact results. I must not have full understanding about how the authors performed these calculations. Perhaps it has something to do with how the 5-year Base Case results were annualized for the 1-year budget impact analysis? Some more detail about the calculations would be appreciated here. You could even show the component parts of the calculation in Table 4. Line 369: You need to reiterate that your study focused on patients “at risk of recurrent venous thromboembolism,” not all cancer patients in general Line 371: Sensitivity analyses – plural, because you performed multiple types of sensitivity analyses Line 374: Probably no need to mention the QALYs number here if you are not mentioning the costs. Besides, 0.012 QALYs over 5 years isn’t particularly impressive to emphasize in the leading paragraph of the discussion. See comment for Line 284. Lines 375-377: The sentence “MB events are . . .” is awkwardly phrased and should be re-written Line 377: Delete “might” – it is clear that it is the only explanation in context of the model Line 381: Do you mean “€161 to €184”? Line 393: Recommend not bringing nadroparin into the discussion or going off on the tangent of comparing results to LMWHs – it only confuses things. Line 409: “they” refers to GI-tumor patients, not DOACs – re-write to clarify Line 430: Mention that these results favor rivaroxaban Line 432: Delete “over” or replace “for over more than” with “beyond” Line 437: Add “cost” in front of “savings”; consider replacing “increase” with “accrue” Line 448: Consider re-phrasing this sentence: “This results in an overestimation of the safety of LMWH.” You are talking about incidence of particular clinical events, not the overall safety of LMWH Line 462: Replace “increasing” with “improving” Line 465: It is a little confusing that you call them “drug costs” here but call them “treatment costs” everywhere else in the paper Line 465: Again, the word “invasive” isn’t the best choice here (also see Line 52) Lines 466-467: Last sentence is clunky; suggestion to just delete it Table S3, Line 27: Not clear how a fatal event can have a utility > 0. Also, where is the utility value for a fatal MB event? Additional General Comments In numerous places throughout the text, you use the term “compared to” when you should be using “compared with”; always use “compared with” when juxtaposing two or more items to illustrate similarities and/or differences (“compared to” is reserved for asserting that two items are similar). The authors should explain better why dalteparin is the targeted comparator for the analysis, without
--	--

	mention of other LMWHs, such as enoxaparin or tinzaparin. Further, the authors should explain the rationale for the class comparison of a DOAC vs. a LMWH. Numerous other therapies were excluded from this analysis, which might be perfectly acceptable, but there should be explanation about this.
--	---

VERSION 1 – AUTHOR RESPONSE

Reviewer: 1

Reviewer Name: Annie Young

Institution and Country: The University of Warwick, United Kingdom

Please state any competing interests or state ‘None declared’: Competing Interest: I am the principal investigator for the study on which this analysis was initially based.

Please leave your comments for the authors below

- *This is a health economic modelling paper, exploring cost effectiveness and budget impact of rivaroxaban and low molecular weight heparin in the cancer population at risk of recurrent VTE, hitherto not carried out using this approach, although this paper needs to be put into the context of recent publication from Strieff et al, J Med Econ. 2019 Nov;22(11):1134-1140, to be strengthen the case.*

We have now discussed the findings of Streiff et al. in the discussion section. Please see Line 453.

General Comments:

I am a clinician and therefore commenting on the clinical aspects of the paper. In this respect, I feel it would be beneficial to have further input from clinical experts on cancer and thrombosis – mainly to update/correct some of the text and include more recent references.

- *Bayer, the makers of rivaroxaban, funded the study and there may be a potential conflict; I do not see a statement on the role of the funder in the identification, design etc of the analysis as in Item 23 of CHEERS checklist, which would indicate no conflict or conflict.*

We do agree that this should be stated more clearly. We have now added a sentence in the funding statement (Line 73) explaining the level of involvement of Bayer in this analysis.

The costs assumptions look good and are applicable to the Netherlands and the scenarios. I welcomed scenario 5 in particular as treatment duration of 6 months does not generally reflect practice in this population, as the authors discuss.

- *The treatment costs which drive the cost benefit analysis may differ in other countries; although implicit in the paper, this should be highlighted.*

We have now mentioned explicitly in the discussion section that the cost-savings were mainly driven by the difference in treatment costs, and that this is specifically the case for the Netherlands and that this may differ in other countries. Please see Line 448.

Specific Comments:

- *Abstract and conclusion: ‘...many cancer patients can benefit...’ I would tone that down to ‘certain cancer patients’, due to clinical challenges which the authors address in the discussion*

Reviewer 2 also had comments on this sentence, and was therefore removed from the abstract. We did adjust this in the conclusion section, please see Line 564.

-
-
- *References for Guidelines on treatment and prevention of recurrence of VTE in the cancer population are out of date (refs 4 and 7-9) so more recent guidelines e.g. ITAC, ASCO update, NCCN update, have different wording in their recommendations as stated in this paper. There appears to be differences, between the introduction and the discussion, in the summary of what the international guidelines state. Lines 96 and 97 are wrong.*

We fully agree that the guidelines we used were outdated. We have now used the ITAC (2019), ASCO update (2019), NCCN update (2018) and the ESC for acute PE (2019) guidelines as references, and adjusted the text in the introduction (Lines 98 – 114), method section (Line 156), and in the discussion section (Lines 442 and 515) accordingly.

- *Line 101 – “Patients with UGI cancers were excluded” – this is inaccurate as these patients were not excluded until nearing the end of the study (ref 44). And select-d did include all patients with GI tumours, in fact the biggest group was patients with colorectal cancer (line 409; reference 14).*

We have now removed this from our manuscript, as the statement was incorrect, and does not add anything to the discussion around the cost-effectiveness analysis (Lines 118 and 497).

- *I appreciate that the model may have been started many months ago; however the 12 month data for the select-d trial I believe were published online in January 2020 and may have provided continuity for the 12 month current state (lines 256, 179 and 435). It would be helpful to state when the analysis was carried out.*

We have stated this now in the methods section, please see Line 145.

- *Discuss the meaning of the estimated health gain of 0.012 QALY with rivaroxaban relative to LMWH e.g. does this fall above or below a threshold for minimally important differences seen in other studies in similar settings?*

This was also one of the comments from the other reviewer (see below). Please see Lines 45, 342, and 445 in the manuscript.

Minor comment:

- *The word data is plural.*
-
-

We have now adjusted this (Line 211).

-
-
- *In conclusion, the model and assumptions made seem reasonable; the health economists have carried out a thorough analysis. However, the clinical rationale for the health economic analyses needs updated and amended. The difference in the budget impact in favour of rivaroxaban is relevant for the Netherlands.*

We have now adjusted the clinical rationale for the analysis by using the latest versions of the guidelines (above mentioned) and we have referred to the article from Streiff et al. from 2019 (also mentioned above) as well as some recent trials assessing the effectiveness and safety of apixaban in patients with cancer who are at risk of recurrent VTE (Line 549).

Reviewer: 2

Reviewer Name: Jeffrey D. Miller

Institution and Country:

IBM Watson Health

Cambridge, Massachusetts, USA

Please state any competing interests or state 'None declared': None declared

Please leave your comments for the authors below

Please see my detailed comments with line references in the attached file.

This is a well-written paper with solid underpinnings of data and methods. All the criteria for good modeling practices were met or exceeded. I prepared a long list of comments and suggested edits which I hope the authors do not find too daunting. Most of the criticisms are relatively minor, and I do not think new analyses or substantial re-writing needs to be undertaken. There was a flawed assumption about "negligible" productivity losses because average patient age in the model is 67 years, but this was unlikely to have any major impact on the analysis results. Some minor re-work should quickly fix this. I do suggest that the authors focus on the fact that this specifically is a rivaroxaban vs. dalteparin analysis, and not try to step out those bounds into portraying this as a DOAC vs. LMWH class analysis. Further, this is an analysis of secondary prevention or prophylaxis of recurrent VTE, not treatment of VTE. The authors are a bit inconsistent with their terms and unintentionally entangle "prevention" and "treatment," but only in their choice of words. Nonetheless, care should be taken to make the distinction. I think some editorial polish will set these and most other things straight.

I extend my congratulations to the authors for a study well done.

We have summarized some minor/textual changes as a response to the reviewer's comments in this table. Some comments that required more explanation are stipulated below.

Comment	Line
Line 43: State who or what organization bears the “financial consequences”	43
Line 44: Replace “appeared to be” with “were” – your analysis findings are observationally concrete and not interpretive (also see Line 283, where this same comment applies)	44
Lines 52-53: This is the first place you mention “direct oral anticoagulant”; you should state earlier that rivaroxaban is in the class of DOAC anticoagulants.	54
Line 59: Probably should add “Markov” to “tunnel states” – i.e., “Markov tunnel states”	61
Line 65: Seems odd to mention apixaban here	68
Line 69: Be more specific about the Bayer Pharma funding – global or a local affiliate?	73
Line 82: Replace “prophylaxis” with “prophylaxis for recurrence”	88
Line 93: Provide examples of DOACs. Isn't this where you want to introduce rivaroxaban as a DOAC?	91
Line 100: It comes later on Lines 132-133, but suggestion to move the explanation of “incidental PE” to here – many readers will not know	117
Line 101: Replace “they” with “the trial researchers”	119
Line 107: Replace “an increase in the use” with “greater utilization”	126
Line 111: Use your acronym VTE for “venous thromboembolism”	130
Line 124: The focus of the analysis is on rivaroxaban in a very specific indication; here it sounds like you are broadly evaluating rivaroxaban in all uses	145
Line 129: State what the software platform was. Excel? TreeAge?	152
Line 135: Specifically talk about cancer here and how cancer aligns with the SELECT-D protocol. Otherwise, readers will wonder “what kind of cancer?”	158, 159
Line 137: Break into two sentences	169
Line 142: Need to state how long “one cycle” is. It comes later on Lines 170-171, but it would benefit to mention here	168

Line 170: Delete “state per cycle. The cycle length was one month.” and replace with “health state per one-month cycle.”	196
Line 173: Replace “the patient” with “patients”	205
Line 179: Replace “this data” with “these data”	211
Line 183: Remind the reader that you are talking about data abstracted from the SELECT-D trial	217
Line 191: Need to state somewhere in this section (perhaps the first sentence) that all costs are standardized to 2019 Euros (as stated in Table S2)	230
Lines 194-195: You state “We assumed no event-related healthcare costs for patients with incidental PE.” Why is this? Further explanation is needed.	234
Line 224: Mention in this section what the utilities are for – i.e., for calculating QALYs! The reasoning behind this won’t be obvious to many readers.	272
Line 230 (and Table S3): Unclear how CTEPH beyond 5 years is relevant for this model.	277
Line 231: Replace “six” with “6”	278
Line 232: Not correct to say that utilities were discounted, because they were not – it is the QALYs calculated from the utilities that were discounted at 1.5% per annum	280
Line 237: Change to: “check the robustness of the model results to uncertainty and known variations in key input parameters”	286
Line 242: You state that there are 15 parameters . . . I see only 12 in the tornado diagrams	292
Line 246: Suggestion that it might be clearer to the reader to summarize the setup of the scenarios and their assumptions in a table	316
Line 287: “scenarios” plural	316
Lines 298-299: Perhaps you mean that these events have the highest incremental cost or incremental cost differences? Insert “incremental”?	364
Line 310 (Table 2): How come you don’t show a “rolled up” line with the aggregation of event costs? You have lines for Treatment Costs and Indirect Costs, but you are missing the third component of Total Costs.	377
Lines 305-310 (Tables 2 and 3): State the base year for the cost estimates – i.e., “2019 Euros”	371, 377
Line 324: Again, there are only 12 parameters, not 15	395
Lines 332 and 337 (Figures 2 and 3): For context, suggestion to put a point of the deterministic result on the plot as a different color (red?); this gives a nice context about how the probabilistic results are arranged around the deterministic result.	Fig 2 and S1

Line 369: You need to reiterate that your study focused on patients “at risk of recurrent venous thromboembolism,” not all cancer patients in general	442
Line 371: Sensitivity analyses – plural, because you performed multiple types of sensitivity analyses	446
Lines 375-377: The sentence “MB events are . . .” is awkwardly phrased and should be re-written	459
Line 377: Delete “might” – it is clear that it is the only explanation in context of the model	461
Line 381: Do you mean “€161 to €184”?	467
Line 409: “they” refers to GI-tumor patients, not DOACs – re-write to clarify	497
Line 430: Mention that these results favor rivaroxaban	522
Line 432: Delete “over” or replace “for over more than” with “beyond”	526
Line 437: Add “cost” in front of “savings”; consider replacing “increase” with “accrue”	530
Line 448: Consider re-phrasing this sentence: “This results in an overestimation of the safety of LMWH.” You are talking about incidence of particular clinical events, not the overall safety of LMWH	543
Line 462: Replace “increasing” with “improving”	559
Line 465: It is a little confusing that you call them “drug costs” here but call them “treatment costs” everywhere else in the paper	562
Lines 466-467: Last sentence is clunky; suggestion to just delete it	565

-
-
- *Title: State openly in the title that is a “rivaroxaban vs. dalteparin” analysis. Otherwise the reader is led to believe that the analysis is a broad comparison of rivaroxaban against all competing therapies, which is not the scope of this study. Also, make this clear in the text too, as sometimes “dalteparin” seems hidden behind the nomenclature of “LMWH”. It is appropriate to mention once that dalteparin is a LMWH, but thereafter the comparator should be distinctly termed “dalteparin”.*
 - *Line 31: State that the comparison is with dalteparin, not the broad class of LMWH. As mentioned above, it is appropriate to mention once that dalteparin is a LMWH, but thereafter the comparator should be distinctly termed “dalteparin”. If you talk about dalteparin in context of LMWHs, then you need to talk about rivaroxaban in context of DOACs.*
 - *Line 87: As mentioned above, you are singling out dalteparin without much explanation about the rationale.*

- *Line 115: Replace “LMWH” with “dalteparin” (see comments above about how you should always be specific that the comparator is dalteparin)*
- *Line 393: Recommend not bringing nadroparin into the discussion or going off on the tangent of comparing results to LMWHs – it only confuses things.*
- *The authors should explain better why dalteparin is the targeted comparator for the analysis, without mention of other LMWHs, such as enoxaparin or tinzaparin. Further, the authors should explain the rationale for the class comparison of a DOAC vs. a LMWH. Numerous other therapies were excluded from this analysis, which might be perfectly acceptable, but there should be explanation about this.*

We referred to the drug class LMWH as the comparator, while the clinical data and all cost inputs were based on the LMWH dalteparin. We fully agree that it is more appropriate to mention once in the introduction that dalteparin is a LMWH, and refer to dalteparin as the comparator throughout the rest of the manuscript. We have now openly stated in the title that this analysis has been done versus dalteparin, and refer consistently throughout the manuscript to dalteparin as the comparator. We have also deleted the part where we brought nadroparin into the discussion (see Line 476).

-
-
- *Line 47: The “€9,834,144” is an annual average, given that you are looking at time horizons both longer and shorter than one year.*
 - *Lines 353 and 362: Using the available information presented in the tables, I was not able to replicate the calculations that yielded these budget impact results. I must not have full understanding about how the authors performed these calculations. Perhaps it has something to do with how the 5-year Base Case results were annualized for the 1-year budget impact analysis? Some more detail about the calculations would be appreciated here. You could even show the component parts of the calculation in Table 4.*

Thank you for the comments. We agree that this can be a confusing for the reader, therefore we have added to the sentences in Line 49, 314, and 418 that this calculation is based on a one-year time horizon. And added an explanation on how this was done in Line 317.

-
-
- *Line 52: Mention of “less invasive” is strange here; perhaps another word choice would be appropriate.*
 - *Line 465: Again, the word “invasive” isn’t the best choice here (also see Line 52)*

Due to another comment we have now removed this specific sentence (Line 54), however, we do still mention it in the conclusion at the end of the manuscript (Line 563). We have changed it into: ‘... its oral administration is more convenient than daily subcutaneous injection’.

-
-
- *Lines 52-53: The conclusion falls flat, as the authors step out of the context of the study, which is a financial analysis, and into the vague area of patient benefit. Isn't the healthcare payer the focus stakeholder in this study?*
 - *Abstract and conclusion: '....many cancer patients can benefit....' I would tone that down to 'certain cancer patients', due to clinical challenges which the authors address in the discussion*

We agree that this sentence is out of the scope of this study. We used a societal perspective; however, this sentence may suggest we use a patient's perspective which is not the case. We have deleted this sentence in the abstract (Line 54) and conclusion (Line 563) of the manuscript.

-
-
- *Lines 138-139: Need to mention in the Discussion that 58% of patients having metastatic cancer might be a limitation of the analysis. Your results and conclusions pertain mostly to severely ill patients. Also, the majority (53%) of patients have incidental PE – perhaps another limitation?*

We have now added further clarification in the discussion section (Line 509).

-
-
- *Line 144: State that these are "Markov tunnel states" and explain what that means; typical readers will not know what a tunnel state is and what it does*
 - *Lines 148-149: Suggest to explain "modeled in the background" – state that incidence and costs of the complications are included in the calculations*

We agree these terms can be difficult to understand for readers who are not experience in health economics. Therefore, we have added some more explanation about the use of tunnel states and 'in background' calculations. Please see Lines 169-175.

- *Lines 171-172: Need to reconcile how you have 6-month transition probabilities for 1-month Markov cycles. Probably just need to explain this a bit.*

We do agree that this part was not clearly explained and have now added the formula that was used to translate event rates into transition probabilities. Please see Line 198.

- *Lines 181-182: This sentence was difficult to understand and needs to be re-written for better clarity: “The distributions of the types of VTE and MB were calculated based on the total number of events and assumed to be treatment-independent, since the total number of events was low.”*

Thank you for your comment. We have adjusted this sentence to: The distributions among the different types of VTE (incidental PE, symptomatic PE, DVT, and fatal PE) and MB (ICH, non-ICH, fatal MB) were calculated based on the total number of events in both arms (rivaroxaban and dalteparin) together and assumed it to be treatment-independent, since the total number of events in the trials was low. Please see Line 213.

- *Line 187: Suggest to explain why “beta distributions” are the best choice – reference to the Briggs paper (#16)?*
- *Line 220: Citation for choice of gamma distribution? The Briggs paper, #16?*

This is indeed explained in the book by Briggs et al., which is now reference number 20 (not #16). We feel it is not within the scope of this study to discuss the reasoning behind the choices of the different distributions as it might get very technical. However, we moved reference 20 right after the choice for distribution and added the reference of the Dutch guideline for economic evaluation of healthcare who also refer to Briggs et al. for the use of distributions for input parameters. We have also added an explanation, please see Line 223.

- *Lines 197-198: You state “Costs of a fatal MB were assumed to be equal to those of non-fatal 198 non-ICH MB.” But aren’t there substantial indirect costs associated with a fatal event? Family expenses, funeral/burial expenses, estate settlement, etc. Table S2 does not show any indirect costs for fatal events.*

We do agree that fatal events are also related with substantial indirect costs and that therefore also informal care costs should be included for the fatal PE and fatal MB events. We have now adjusted this in the model and corrected all the results accordingly. We have added a note in Table S2 explaining that informal care costs were applied for fatal as well as non-fatal PE and MB.

- *Lines 205-206: You state “Rivaroxaban users were assumed to require an annual check-up of their renal function.” Cancer patients (especially those with metastatic disease) are at higher risk for renal impairment and are tested much more frequently. Some investigation into the literature would be revealing here. This omission probably is inconsequential for the model results, but should be checked.*

We have now added some more explanation on this in the discussion section, please see line 479.

- *Line 206: What about the cost for a healthcare worker to teach the patient how to self-inject? There is a one-time cost for that. I see in Lines 388-389 you state that you were “conservative” in not including such costs, but it seems more like an omission rather than a safe assumption.*

We have now included the costs related to the education of self-injection of dalteparin in the base case and all other scenarios in the model. Please see Line 246 and Table S2. We have removed the line in the discussion stating that this was a conservative assumption. All results were revised.

- *Line 216: You state “productivity losses were assumed to be negligible” because the average patient age was 67 years. However, this logic is flawed because not every patient in the model is exactly 67 years old. Instead, you are essentially modeling patients in a wide distribution of ages, many (but maybe not most) who are of working age and suffer work/productivity impediments. As an illustrative example, the average age of a 55 year old worker and a 79 year old retiree is 67 years.*

We do agree that productivity losses might not necessarily be negligible, however, there is still no data substantiating the effect on productivity losses of a VTE or anticoagulation-related bleeding event in patients with cancer. It is likely that these events do not have the same effect on productivity losses as they do in the general population experiencing these events, as these patients are already severely ill. Moreover, the average age from the SELECT-D trial was used as the starting age in the model and the fact the majority (58%) of the patients in the SELECT-D trial had metastatic cancer may indicate a low employment rate. This is now better explained in the methods section (Line 262).

-
-
- *Line 281: Suggestion to spend a little more time here explaining what is going on with the results with regard to “economic dominance.” You need to explain that that rivaroxaban always simultaneously confers better clinical and quality-of-life outcomes at less cost. As such, a numerical ICER is not presented because it has no meaning (hence, why you say “Dominant” in the far-right column of Table 2). This is all perfectly understandable to health economists, but the general readership of the journal is going to be lost without some education about this.*

Thank you for this suggestion. We have now added your suggestions for a better explanation in Line 340.

-
-
- *Line 284: 0.012 QALYs is equivalent to 4.4 quality-adjusted life days, and that is over 5 years (so about 21 hours per year!). When expressed at this level, it isn’t a particularly impressive figure. In fact, it is almost negligible. As such, rivaroxaban and dalteparin are basically equivalent with regard to promoting quality of life. The authors might want to address this somewhere in the paper, perhaps in the discussion section.*
 - *Line 374: Probably no need to mention the QALYs number here if you are not mentioning the costs. Besides, 0.012 QALYs over 5 years isn’t particularly impressive to emphasize in the leading paragraph of the discussion. See comment for Line 284.*

We have now added a sentence stressing the fact that the increase in QALYs is a very marginal increase, with the example that it is equivalent to 4.4 quality-adjusted life days over five years’ time. Please see Line 344.

-
-
- *Line 295: Not clear why you look only at Base Case and Scenario 4 here and in Table 3. What about the other scenarios?*
 - *Line 323: Should you mention in the text that the tornado diagrams are for the Base Case analyses? You do state this in the caption for the diagrams, but probably best to mention in the text too. Also for consideration, you showed probabilistic sensitivity analysis results for Scenario 4, but you chose not to show univariate sensitivity analysis results for Scenario 4? Perhaps show the tornado diagrams for Scenario 4 in the Supplement? Maybe the Scenario 4 scatterplot diagram (Figure 3) should go in the Supplement too, and just show the Base Case results in the main text?*
 - *Line 262: This isn’t clear enough to fully understand: “drug-specific distributions of the types of VTE and MB.”*

We have now summarized the scenarios in Table 2 and better explained why we included scenario 6 with the drug-specific distributions of the types of VTE and MB (see Line 316). Also, we have now

changed the order of the scenarios, so that it makes more sense to show only the results from the base case and scenario analysis 1 in Table 4.

- *Lines 318-319: You state “77.0% were located in the south-eastern quadrant,” but that isn’t going to have much meaning to the average reader; suggestion to better explain the meaning and implications*
- *Lines 319-320: You state “22.8% are considered cost-saving but less effective compared to LMWH”; given that 1 of every 5 points is in this southwest quadrant, suggestion to better explain the meaning and implications of results that fall into this quadrant, especially how to interpret the nebulous notion of “decremental cost-effectiveness”*

Thank you for this comment. We have now removed the term ‘southwest quadrant’ and explained that rivaroxaban was in 77.8% (base case) and 98.8% (scenario 4) of the iterations in the probabilistic sensitivity analysis cost-saving and more effective compared with dalteparin. Please see Line 385 for the adjustments.

- *Table S3, Line 27: Not clear how a fatal event can have a utility > 0. Also, where is the utility value for a fatal MB event?*

We fully agree, so we have now assumed that all the fatal events have a utility of 0. Therefore, we had to recalculate the results, which are all corrected in the manuscript.

- *In numerous places throughout the text, you use the term “compared to” when you should be using “compared with”; always use “compared with” when juxtaposing two or more items to illustrate similarities and/or differences (“compared to” is reserved for asserting that two items are similar).*

Thank you for this comment, we have now adjusted ‘compared to’ to ‘compared with’ throughout the whole manuscript.

VERSION 2 – REVIEW

REVIEWER	Annie Young University of Warwick, UK
REVIEW RETURNED	31-Aug-2020

GENERAL COMMENTS	Excellent changes, based on an extremely detailed review by reviewer 2. My comments all addressed satisfactorily. Thank you. 1. New comment: I feel recurrent VTE and major bleeding should be mentioned in the abstract and more specific discussion of cost of recurrent VTE and the cost of major bleeding (the 'balance' as the authors say) should be had, based on the results. 2. data are - still missing in many places, although this has been changed to these! I am happy for the editors to check these two points Well done on good work and now, good manuscript.
--

REVIEWER	Jeffrey D. Miller IBM Watson Health, USA
REVIEW RETURNED	20-Aug-2020

GENERAL COMMENTS	This revision is much improved and worthy of publication. Please see the attached PDF with my remaining comments, most of which are just minor issues or suggested corrections. One critical revision still remaining (which I neglected to mention in the first draft) is that all the tornado diagrams are incomplete because the numerical range endpoints are not depicted and the legends for the color bars are absent. In my PDF comments, I reference a publication with an excellent example of a tornado diagram, and it will not take much additional work to bring your diagrams up to this level of quality.
--

VERSION 2 – AUTHOR RESPONSE

Reviewer(s)' Comments to Author:

Reviewer: 2

Reviewer Name

Jeffrey D. Miller

Institution and Country

IBM Watson Health, USA

Please state any competing interests or state 'None declared':

None

Please leave your comments for the authors below

This revision is much improved and worthy of publication. Please see the attached PDF with my remaining comments, most of which are just minor issues or suggested corrections. One critical revision still remaining (which I neglected to mention in the first draft) is that all the tornado diagrams

are incomplete because the numerical range endpoints are not depicted and the legends for the color bars are absent. In my PDF comments, I reference a publication with an excellent example of a tornado diagram, and it will not take much additional work to bring your diagrams up to this level of quality.

Dear reviewer,

Thank you for the comments. We have incorporated all your remaining comments in the manuscript, and adjusted the tornado diagrams according to the example.

Reviewer: 1

Reviewer Name

Annie Young

Institution and Country

University of Warwick, UK

Please state any competing interests or state 'None declared':

I was principal investigator for the study on which the meta-analysis was based.

Please leave your comments for the authors below

Excellent changes, based on an extremely detailed review by reviewer 2.

My comments all addressed satisfactorily. Thank you.

1. New comment: I feel recurrent VTE and major bleeding should be mentioned in the abstract and more specific discussion of cost of recurrent VTE and the cost of major bleeding (the 'balance' as the authors say) should be had, based on the results.
2. data are - still missing in many places, although this has been changed to these!

I am happy for the editors to check these two points

Well done on good work and now, good manuscript.

Dear reviewer,

Thank you for the comments. We apologise for missing out on some of the corrections. We have now adjusted all 'data is' to 'data are'. With regard to the first comment: we have now mentioned the balance between VTE recurrence and bleeding in the abstract (rows 36-37) and discussion (rows 404-425).

VERSION 3 – REVIEW

REVIEWER	Jeffrey D. Miller IBM Watson Health
REVIEW RETURNED	26-Oct-2020
GENERAL COMMENTS	I recommend acceptance; no further comments.